# Mendelian randomization analysis provides causality of smoking on the expression of ACE2, a putative SARS-CoV-2 receptor

**Hui Liu[1], Junyi Xin[2], Sheng Cai[3], Xia Jiang[4]***

[1]Biomedical Research Center, Zhejiang Provincial Key Laboratory of Laparoscopic Technology, Sir Run Run Shaw Hospital, School of Medicine, Zhejiang University, Hangzhou, China; [2]Department of Environmental Genomics, Jiangsu Key Laboratory of Cancer Biomarkers, Prevention and Treatment, Collaborative Innovation Center for Cancer Personalized Medicine, School of Public Health, Nanjing Medical University, Nanjing, China; [3]Institute of Drug Metabolism and Pharmaceutical Analysis, Zhejiang Province Key Laboratory of Anti-Cancer Drug Research, Zhejiang University, Hangzhou, China; [4]Department of Clinical Neuroscience, Center for Molecular Medicine, Karolinska Institute, Stockholm, Sweden

## Abstract

**Background:** To understand a causal role of modifiable lifestyle factors in angiotensin-converting enzyme 2 (ACE2) expression (a putative severe acute respiratory syndrome coronavirus 2 [SARS-CoV-2] receptor) across 44 human tissues/organs, and in coronavirus disease 2019 (COVID-19) susceptibility and severity, we conducted a phenome-wide two-sample Mendelian randomization (MR) study.

**Methods:** More than 500 genetic variants were used as instrumental variables to predict smoking and alcohol consumption. Inverse-variance weighted approach was adopted as the primary method to estimate a causal association, while MR-Egger regression, weighted median, and MR pleiotropy residual sum and outlier (MR-PRESSO) were performed to identify potential horizontal pleiotropy.

**Results:** We found that genetically predicted smoking intensity significantly increased ACE2 expression in thyroid ($\beta$=1.468, p=1.8×10$^{-8}$), and increased ACE2 expression in adipose, brain, colon, and liver with nominal significance. Additionally, genetically predicted smoking initiation significantly increased the risk of COVID-19 onset (odds ratio=1.14, p=8.7×10$^{-5}$). No statistically significant result was observed for alcohol consumption.

**Conclusions:** Our work demonstrates an important role of smoking, measured by both status and intensity, in the susceptibility to COVID-19.

**Funding:** XJ is supported by research grants from the Swedish Research Council (VR-2018–02247) and Swedish Research Council for Health, Working Life and Welfare (FORTE-2020–00884).

*For correspondence: xia.jiang@ki.se

**Competing interests:** The authors declare that no competing interests exist.

## Introduction

The severe acute respiratory syndrome coronavirus 2 (SARS-CoV-2) has led to a worldwide pandemic of coronavirus disease 2019 (COVID-19) (*Coronaviridae Study Group of the International Committee on Taxonomy of Viruses, 2020*; *World Health Organization, 2020*). As a host receptor of SARS-CoV-2, the expression level of angiotensin-converting enzyme 2 (ACE2) has been found to influence both the risk and severity of infection (*Hoffmann et al., 2020*; *Wrapp et al., 2020*; *Zhou et al., 2020a*; *Li et al., 2003*; *Li et al., 2005*). Moreover, a growing body of evidence from

epidemiological investigations has demonstrated a substantial disparity in the susceptibility to infection (*Guan et al., 2020*; *Hu et al., 2020*; *Mehra et al., 2020*; *Patanavanich and Glantz, 2020*). For example, a multi-center study involving 8910 COVID-19 cases from 169 hospitals in Europe and North America identified an increased risk of in-hospital death among current smokers (odds ratio [OR]=1.79; 95% CI: 1.29–2.47) compared with ever-smokers or non-smokers (*Mehra et al., 2020*).

Consistent with findings from large-scale population-based observational studies, a laboratory-based study involving 131 RNA-sequenced human lung cancer tissues (54 samples of European ancestry individuals and 77 samples of Asian ancestry individuals) found that smokers expressed a significantly higher level of ACE2 compared to non-smokers in both populations, leading to a potentially heightened susceptibility to SARS-CoV-2 infection (*Cai, 2020*). Furthermore, incorporating two additional DNA microarray datasets of lung cancer, the significant smoking-ACE2 association observed in a total of 224 samples did not alter after adjusting for age, sex, race, and platforms. Nevertheless, these samples are derived from lung cancer patients, restricting its generalizability to normal lung tissues and to the general population. In a related work, Rao et al. conducted a phenome-wide Mendelian randomization (MR) study examining an extensive amount of diseases, traits, and blood proteins and identified several 'exposures' including diabetes, breast cancer, lung cancer, inflammatory bowel disease, and smoking to increase ACE2 expression in normal lung tissue (*Rao et al., 2020*). This analysis, despite its substantially augmented number of exposures (N=3948), has several limitations. First of all, disease status such as diabetes and cancers are difficult to modify, at the population level it is more important to discover and intervene with modifiable risk factors such as smoking and alcohol consumption. However, regarding smoking, only three single nucleotide polymorphisms (SNPs) were used as instrumental variables (IVs) by Rao et al., which explained negligible phenotypic variation and did not accurately predict smoking status. The hitherto largest genome-wide association study (GWAS) of tobacco use was conducted in a total of 1.2 million individuals and discovered over 400 genetic variants associated with smoking initiation and intensity (*Liu et al., 2019*). Last but not least, Rao et al. focused on lung tissue instead of systemically examining all human tissues. Despite lungs being the most relevant and vulnerable organ to a respiratory syndrome COVID-19, recent studies have identified the involvement of other human tissues (e.g. gastrointestinal tract) in SARS-CoV-2 infection (*Zou et al., 2020*).

Motivated by these findings, we aim to explore whether genetic predisposition to common human modifiable behaviours including smoking and alcohol consumption could lead to an increased ACE2 expression, which subsequently yields to an increased susceptibility and severity of COVID-19. Here, we conduct a phenome-wide MR analysis by incorporating ACE2 expressions from a broad spectrum of tissues/organs available in the GTEx database. As one of the hitherto largest databases with concomitant information on DNA genotype and RNA expression, GTEx collects a large variety of tissues (N=44) from healthy population (deceased donors) (*Gamazon et al., 2018*). In addition, data on COVID-19 susceptibility and severity were obtained from COVID-19 Host Genetics Initiative, a global project aims to understand the role of host genome in COVID-19 outcome (*COVID-19 Host Genetics Initiative, 2020*). Hundreds of genetic variants identified by a large-scale GWAS of tobacco use and alcohol consumption were used as IVs – incorporating additional loci greatly enhances the strength of genetic instruments as well as both accuracy and precision of MR estimates (*Liu et al., 2019*).

## Materials and methods

### Data on IV-exposure

IV-exposure associations were extracted from the hitherto largest GWAS conducted by the GCSCAN consortium (GWAS and Sequencing Consortium of Alcohol and Nicotine use) for tobacco use and alcohol consumption, totalling 1.2 million individuals of European ancestry (*Liu et al., 2019*). This GWAS firstly meta-analysed summary-level data from each participating cohort and identified independent SNPs passing genome-wide significance ($p<5\times10^{-8}$) based on linkage disequilibrium. After that, additional independent and genome-wide significant SNPs were selected using a conditional analysis within each significant locus defined as a 1 MB region surrounding the sentinel variant (the variant in the locus with the lowest p-value). We used all conditionally independent biallelic SNPs as IVs.

In our analysis, we included two smoking phenotypes and one drinking phenotype, smoking initiation as reflected by never vs. ever smoking (IV=378, N=1,232,091), smoking intensity as reflected by cigarettes per day (IV=55, N=337,334), and common (opposing to excessive or harmful) alcohol drinking behaviour defined as drinks per week (IV=99, N=941,280). The proportion of phenotypic variance explained by IVs accounted for 2.3% for smoking initiation, 1.1% for cigarettes per day, and 0.2% for drinks per week. Detailed information regarding IVs for each exposure were shown in *Supplementary file 1a-1c*.

Strong instrumental variable is the basic requirement to ensure a valid MR result. The strength of IV was verified by calculating F-statistics using the formula $F = \frac{R^2(n-1-k)}{(1-R^2)k}$, where $R^2$ is the proportion of variance explained by the IV, k refers to the number of IVs, and n indicates the sample size (*Pierce et al., 2011*). The F-statistics for smoking initiation, smoking intensity (cigarettes per day), and alcohol consumption (drinks per week) were 77.2, 67.4, and 17.8, respectively, indicating strong IVs (F-statistics > 10) for each of our exposure of interest.

## Data on IV-outcome

Associations of genetic variants with ACE2 expression were extracted from the GTEx database release version 8 available at the GTEx Portal (http://www.gtexportal.org). It is one of the largest databases with concomitant information on genotype and expression data for a large variety of non-diseased tissues collected from >1000 human donors. Out of the total 54 tissues/organs, we focused on ACE2 expression in 44 tissues/organs with a decent sample size involving at least 100 individuals to ensure statistical power (*Supplementary file 1d*). Specifically, the associations of genotype with ACE2 expression in adipose tissue, artery, brain, colon, oesophagus, heart, liver, lung, minor salivary gland, muscle, nerve, ovary, pancreas, pituitary, prostate, skin, small intestine, stomach, testis, thyroid, uterus, and vagina were included.

Unlike most existing MR studies that consider disease status as outcomes, this MR study treats gene expression levels as outcomes. GTEx identifies expression quantitative trait loci (eQTL) by associating genetic variations called from GWAS with gene expression levels obtained from RNA-sequencing. Expression values for each gene were inverse quantile normalized to a standard normal distribution across samples as previously described (*Gamazon et al., 2018*). Please note, here we studied ACE2 RNA expression instead of its actual protein expression. Since ACE2 RNA and protein quantities have been found to be well correlated in tissues such as lung and kidney (*Wang et al., 2020*), ACE2 RNA expression acts as a good proxy for ACE2 protein expression.

In addition to ACE2 expression, associations of genetic variants with COVID-19 susceptibility and severity were extracted from the COVID-19 Host Genetics Initiative (https://www.covid19hg.org). It is a global genetics collaboration aiming to explore the genetic determinants of COVID-19 susceptibility and severity. We used the summary statistics from the data freeze 5 of GWAS meta-analysis, which was released publicly on January 18, 2021. Three COVID-19 outcomes, including susceptibility to COVID-19, hospitalized COVID-19, and very severe respiratory confirmed COVID-19, were used in our analysis.

Briefly, susceptibility to COVID-19 cases was defined as individuals with laboratory-confirmed positive for SARS-Cov-2 infection (via nucleic acid amplification test or serological test), clinician diagnosis, health record evidence by ICD coding, or self-reported ($N_{case}$=38,984 vs. $N_{control}$=1,644,784). Hospitalized COVID-19 cases were defined as individuals hospitalized due to COVID-19 related symptoms with laboratory-confirmed positive for SARS-Cov-2 infection ($N_{case}$=9986 vs. $N_{control}$=1,877,672). Very severe respiratory confirmed COVID-19 cases were defined as hospitalized individuals with laboratory-confirmed positive for SARS-Cov-2 infection, who needed respiratory support except for simple oxygen supplementary or died due to COVID-19 ($N_{case}$=5101 vs. $N_{control}$=1,383,241).

## Statistical analysis

MR uses SNPs as proxies for exposure(s) assuming that SNPs are randomly allotted at conception mirroring a randomized procedure and that SNPs always precede disease onset to eliminate reverse causality. Three essential model assumptions need to be fulfilled to guarantee valid IVs (*Zheng et al., 2017*), that is, IVs are associated with the exposure (relevance assumption); there is no association between IVs and any confounders of the exposure-outcome relationship (independence

assumption); and IVs are associated with the outcome only through the studied exposure (exclusion restriction assumption). If all three model assumptions are satisfied, a causal relationship can be made based on the observed IV-exposure and IV-outcome associations.

We conducted a two-sample MR, where IV-exposure and IV-outcome associations were estimated in two non-overlapping samples. The inverse-variance weighted (IVW) approach was applied as the primary method to estimate the causal link between exposures (smoking and alcohol consumption) and outcomes (ACE2 expression and COVID-19 related adverse outcomes) (*Burgess et al., 2015*). The causal estimate is calculated as a ratio of which the IV-outcome association was divided by the IV-exposure association for each IV and combined across multiple IVs weighted by the reciprocal of an approximate expression for their asymptotic variance. To evaluate potential heterogeneity among causal effects of different genetic variants, Cochran's Q test was performed and $p < 0.05$ was considered as the existence of heterogeneity (*Greco M et al., 2015*).

One major concern of MR is horizontal pleiotropy, meaning genetic variants influence the outcome other than through the exposure. A series of sensitivity analyses were conducted to detect and correct for such a scenario. First, MR-Egger regression was adopted to examine the presence of potential pleiotropy, as well as to complement results from main analysis (IVW) (*Bowden et al., 2015*). When the instrument strength independent of direct effect assumption holds, intercept of MR-Egger regression that differs from zero indicates the presence of horizontal pleiotropy. In addition to the IVW approach and the MR-Egger regression, the weighted median method and the MR-pleiotropy residual sum and outlier (MR-PRESSO) test were also used to detect potential horizontal pleiotropy. The beta estimate of weighted median is calculated as the median of the weighted empirical distribution function of the ratio IV estimates evaluated using each genetic variant individually (*Bowden et al., 2016*). The p-value of MR-PRESSO global test less than $1.0 \times 10^{-6}$ indicates the presence of horizontal pleiotropy. MR-PRESSO not only identifies horizontal pleiotropy by comparing the observed residual sum of squares to the expected residual sum of squares, but also corrects for horizontal pleiotropy through removing outliers (*Verbanck et al., 2018*). Second, we excluded palindromic SNPs with ambiguous strand identification and performed the IVW method on the remaining SNPs. Subsequently, we removed SNPs associated with potential confounding traits as confirmed by the GWAS Catalog (basically any other trait than our exposure of interest). Moreover, leave-one-out analysis was performed where we excluded one SNP at a time and conducted IVW on the rest SNPs to identify the potential influence of outlying variants on the overall estimates. Results were considered significant only if they passed statistical significance ($p < 0.05$) in the IVW approach or the MR-Egger regression and remained directional consistent in the weighted median and MR-PRESSO methods across both primary and sensitivity analyses. We additionally set a corrected p threshold as dividing 0.05 by the number of outcomes $0.05/(44+3) = 1.0 \times 10^{-3}$ (including 44 tissues/organs and 3 COVID-19 outcomes).

Finally, the power of current study was estimated according to a method suggested by *Brion et al., 2013*. To guarantee statistical power, we only included tissues/organs with at least 100 samples in the GTEx database. Under the current sample size, given 1–2% of the phenotypic variance of smoking and alcohol consumption explained by IVs, our study had sufficient power (>80%) to detect a causal effect of 0.74–2.66 in ACE2 expression, and to detect an OR ranging from 1.11 to 1.39 for COVID-19 related outcomes (*Supplementary file 1e*).

## Results

We extracted a total of 532 independent SNPs that achieved genome-wide significance for smoking initiation (N=378), cigarettes per day (N=55), and drinks per week (N=99) from the GSCAN consortium and used as IVs (*Supplementary file 1a-c*). We were able to map 80–95% of those IVs to the GTEx database and to the COVID-19 Host Genetics Initiative database – a virtually complete coverage. The flowchart of IV-selection is shown in *Figure 1*.

We found that genetically instrumented smoking initiation was associated with a significantly increased ACE2 expression in brain putamen basal ganglia (β=1.117, p=0.006), in brain hypothalamus (β=0.848, p=0.022), and in subcutaneous adipose tissue (β=0.285, p=0.016) using the IVW approach (*Table 1*). Results remained directional consistent in the MR-Egger regression although a slightly increased statistical uncertainty was observed (β=1.667, p=0.334 in brain putamen basal ganglia; β=0.963, p=0.545 in brain hypothalamus; β=0.663, p=0.205 in subcutaneous adipose tissue).

**GWAS & Sequencing Consortium of Alcohol and Nicotine use (GSCAN)**
- Smoking initiation: never vs. ever smoking.
- Cigarettes per day: to measure the heaviness of smoking.
- Drinks per week: to measure alcohol use.

The primary focus of the original GWAS was to test variants with minor allele frequency (MAF) ≥1%, as these will be imputed with high confidence. The statistical significance threshold was set as $5.0 \times 10^{-8}$.

**Independent SNPs** that were associated with each trait **passing the significance threshold** were selected as instruments.

**After meta-GWAS**
- 378 SNPs were associated with smoking initiation
- 55 SNPs were associated cigarette per day
- 99 SNPs were associated with drinks per week

**In our MR analysis, these instruments were further removed**
- SNPs that were not matched to the outcome (ACE2 tissue expression and COVID19 related outcomes)
- SNPs with genotyping call rate < 95%
- SNPs with linkage disequilibrium $r^2 < 0.1$
- *P*-value for Hardy-Weinberg equilibrium test $< 1.0 \times 10^{-6}$

**For our primary MR analysis**
- 351-362 smoking initiation-associated SNPs, 45-49 cigarettes/day-associated SNPs and 84-90 drinks/week-associated SNPs were included.
- Number of instruments differs by different tissue types (our outcome).
- ~80-95% coverage of the original GWAS-identified independent SNPs.

**Figure 1.** Flowchart on the selection of instrumental variables.

Additionally, increased ACE2 expression in two colon tissues was observed only through the MR-Egger regression (transverse colon [β=1.129, p=0.017] and sigmoid colon [β=1.925, p=0.042]) and the direction of estimates remained consistent in the IVW approach. For all these associations, we did not find apparent heterogeneity as indicated by Cochran's Q statistics (all p>0.05) or horizontal pleiotropy as indicted by MR-Egger intercept (all p>0.05) and MR-PRESSO global test (all p>$1.0 \times 10^{-6}$), except that horizontal pleiotropy was observed in transverse colon by MR-Egger intercept (p=0.04). Although significant associations of smoking initiation with ACE2 expression in brain caudate basal ganglia and in cerebellar hemisphere were found using the IVW approach, the direction of estimates was opposite in the MR-Egger regression (*Supplementary file 1f*). These associations were therefore not considered as informative.

**Table 1.** Causal association of smoking initiation and angiotensin-converting enzyme 2 (ACE2) expression.

| Organ/tissue | Method | N of IVs | Beta | SE | p | p* |
|---|---|---|---|---|---|---|
| Adipose – subcutaneous | Inverse-variance weighted | 358 | 0.285 | 0.118 | 0.016 | 0.834 |
| | MR-Egger | 358 | 0.663 | 0.522 | 0.205 | 0.456 |
| | Weighted median | 358 | 0.142 | 0.186 | 0.444 | |
| | MR-PRESSO | 358 | 0.285 | 0.118 | 0.017 | 0.846 |
| Brain – hypothalamus | Inverse-variance weighted | 356 | 0.848 | 0.369 | 0.022 | 0.614 |
| | MR-Egger | 356 | 0.963 | 1.588 | 0.545 | 0.941 |
| | Weighted median | 356 | 0.549 | 0.566 | 0.332 | |
| | MR-PRESSO | 356 | 0.848 | 0.369 | 0.022 | 0.655 |
| Brain – putamen (basal ganglia) | Inverse-variance weighted | 357 | 1.117 | 0.406 | 0.006 | 0.334 |
| | MR-Egger | 357 | 1.667 | 1.724 | 0.334 | 0.743 |
| | Weighted median | 357 | 1.256 | 0.607 | 0.039 | |
| | MR-PRESSO | 357 | 1.117 | 0.406 | 0.006 | 0.321 |
| Colon – sigmoid | Inverse-variance weighted | 359 | 0.314 | 0.214 | 0.143 | 0.887 |
| | MR-Egger | 359 | 1.925 | 0.945 | 0.042 | 0.080 |
| | Weighted median | 359 | 0.473 | 0.334 | 0.156 | |
| | MR-PRESSO | 359 | 0.314 | 0.214 | 0.144 | 0.904 |
| Colon – transverse | Inverse-variance weighted | 359 | 0.193 | 0.113 | 0.088 | 0.348 |
| | MR-Egger | 359 | 1.129 | 0.471 | 0.017 | 0.041 |
| | Weighted median | 359 | 0.262 | 0.165 | 0.114 | |
| | MR-PRESSO | 359 | 0.193 | 0.113 | 0.089 | 0.362 |

*p indicates p-value of heterogenous from inverse-variance weighted (IVW) approach, or p-value of intercept from MR-Egger regression, or p-value from Mendelian randomization pleiotropy residual sum and outlier (MR-PRESSO) global test.

We further identified that genetically predicted smoking intensity as reflected by cigarettes per day was associated with a significantly elevated ACE2 expression in thyroid (β=1.468, p=$1.8\times10^{-8}$), in liver (β=1.216, p=0.009), in brain hypothalamus (β=1.789, p=0.014), and in ovary (β=1.545, p=0.026) using the IVW method (*Table 2*). Results remained directional consistent in the MR-Egger regression (β=1.739, p=0.062 in thyroid; β=1.132, p=0.226 in liver; β=0.041, p=0.983 in brain hypothalamus; β=1.658, p=0.347 in ovary). On the contrary, levels of ACE2 expression decreased with genetically instrumented smoking intensity in sigmoid colon tissue (β=−1.971, p=0.019) and in vagina tissue (β=−3.271, p=0.043) using the MR-Egger regression. For all these associations, no apparent horizontal pleiotropy and heterogeneity was found (*Supplementary file 1g*).

For alcohol consumption defined as drinks per week, only one suggestive association with ACE2 expression in tibial nerve was observed using the IVW approach (β=−1.462, p=0.006). However, the direction of effect from the MR-Egger regression was opposite (β=0.336, p=0.721). We therefore considered an overall null association as our main finding with alcohol consumption (*Supplementary file 1h*).

In the sensitivity analyses where we excluded palindromic or pleiotropic SNPs, results remained largely consistent with our primary findings (full results shown in *Supplementary file 1i-1n*). We sequentially excluded proxy SNPs to identify random error introduced by imperfect proxies. In the leave-one-out analyses where we iteratively removed one SNP each time and performed the IVW approach using the remaining SNPs, results were again concordant with our primary findings, indicating an absence of outlying SNPs (*Appendix 1—figure 1*).

Finally, complementing to findings of ACE2 expression, we tested a putative causal link between smoking status, smoking intensity, alcohol consumption, and the risk of COVID-19 related adverse outcomes. We found that smoking initiation significantly increased the risk of COVID-19 onset (IVW: OR=1.15, 95%CI: 1.07–1.23, p=$8.7\times10^{-5}$) even after taking into account multiple comparisons (*Table 3*). Results remained significant in the weighted median, and MR-PRESSO methods, however,

**Table 2.** Causal association of cigarettes per day and angiotensin-converting enzyme 2 (ACE2) expression.

| Organ/tissue | Method | N of IVs | Beta | SE | p | p* |
|---|---|---|---|---|---|---|
| Brain – hypothalamus | Inverse-variance weighted | 48 | 1.789 | 0.730 | 0.014 | 0.959 |
| | MR-Egger | 48 | 0.041 | 1.920 | 0.983 | 0.310 |
| | Weighted median | 48 | 2.182 | 1.347 | 0.105 | |
| | MR-PRESSO | 48 | 1.789 | 0.730 | 0.018 | 0.964 |
| Colon – sigmoid | Inverse-variance weighted | 47 | −0.832 | 0.439 | 0.058 | 0.652 |
| | MR-Egger | 47 | −1.971 | 0.807 | 0.019 | 0.092 |
| | Weighted median | 47 | −1.182 | 0.747 | 0.114 | |
| | MR-PRESSO | 47 | −0.832 | 0.439 | 0.064 | 0.701 |
| Liver | Inverse-variance weighted | 47 | 1.216 | 0.468 | 0.009 | 0.807 |
| | MR-Egger | 47 | 1.132 | 0.922 | 0.226 | 0.913 |
| | Weighted median | 47 | 1.058 | 0.850 | 0.213 | |
| | MR-PRESSO | 47 | 1.216 | 0.468 | 0.012 | 0.843 |
| Ovary | Inverse-variance weighted | 48 | 1.545 | 0.693 | 0.026 | 0.837 |
| | MR-Egger | 48 | 1.658 | 1.745 | 0.347 | 0.943 |
| | Weighted median | 48 | 2.545 | 1.217 | 0.037 | |
| | MR-PRESSO | 48 | 1.545 | 0.693 | 0.031 | 0.844 |
| Thyroid | Inverse-variance weighted | 47 | 1.468 | 0.392 | $1.8 \times 10^{-4}$ | 0.604 |
| | MR-Egger | 47 | 1.739 | 0.907 | 0.062 | 0.739 |
| | Weighted median | 47 | 1.435 | 0.641 | 0.025 | |
| | MR-PRESSO | 47 | 1.468 | 0.392 | $5.0 \times 10^{-4}$ | 0.670 |
| Vagina | Inverse-variance weighted | 48 | −1.150 | 0.688 | 0.094 | 0.055 |
| | MR-Egger | 48 | −3.271 | 1.574 | 0.043 | 0.142 |
| | Weighted median | 48 | −2.644 | 0.916 | 0.004 | |
| | MR-PRESSO | 48 | −1.150 | 0.688 | 0.101 | 0.057 |

*p indicates p-value of heterogenous from inverse-variance weighted (IVW) approach, or p-value of intercept from MR-Egger regression, or p-value from Mendelian randomization pleiotropy residual sum and outlier (MR-PRESSO) global test.

**Table 3.** Causal link of smoking initiation with the risk of coronavirus disease 2019 (COVID-19) related adverse outcomes.

| Outcome | Method | N of IVs | OR (95% CI) | p | p* |
|---|---|---|---|---|---|
| COVID-19 susceptibility | Inverse-variance weighted | 352 | 1.15 (1.07–1.23) | $8.7 \times 10^{-5}$ | $6.7 \times 10^{-5}$ |
| | MR-Egger | 352 | 1.11 (0.83–1.49) | 0.489 | 0.821 |
| | Weighted median | 352 | 1.18 (1.08–1.29) | $2.9 \times 10^{-4}$ | |
| | MR-PRESSO | 352 | 1.15 (1.07–1.23) | $1.0 \times 10^{-4}$ | $5.0 \times 10^{-5}$ |
| Hospitalized COVID-19 | Inverse-variance weighted | 351 | 1.32 (1.16–1.50) | $3.7 \times 10^{-5}$ | 0.009 |
| | MR-Egger | 351 | 0.78 (0.44–1.36) | 0.383 | 0.059 |
| | Weighted median | 351 | 1.37 (1.14–1.66) | 0.001 | |
| | MR-PRESSO | 351 | 1.32 (1.16–1.50) | $4.6 \times 10^{-5}$ | 0.011 |
| Very severe respiratory confirmed COVID-19 | Inverse-variance weighted | 352 | 1.25 (1.03–1.51) | 0.025 | 0.114 |
| | MR-Egger | 352 | 0.83 (0.37–1.89) | 0.658 | 0.318 |
| | Weighted median | 352 | 1.17 (0.89–1.55) | 0.267 | |
| | MR-PRESSO | 352 | 1.25 (1.03–1.51) | 0.025 | 0.119 |

*p indicates p-value of heterogeneous from inverse-variance weighted (IVW) approach, or p-value of intercept from MR-Egger regression, or p-value from Mendelian randomization pleiotropy residual sum and outlier (MR-PRESSO) global test.

showed larger statistical uncertainties in the MR-Egger regression. In addition, smoking initiation increased the risk of very severe respiratory confirmed COVID-19 and hospitalized COVID-19 using the IVW approach, but the results were not supported by the MR-Egger regression. Smoking intensity (cigarettes per day) only increased the risk of very severe respiratory confirmed COVID-19 as shown in the MR-Egger regression (OR=5.99, 95%CI: 1.57–22.84, p=0.012) (*Supplementary file 1o*). On the contrary, we did not find a causal link between alcohol consumption (drinks per week) and the risk of COVID-19 adverse outcomes (*Supplementary file 1p*). Findings did not alter in the sensitivity analyses (*Supplementary file 1q-r* and *Appendix 1—figure 4*).

## Discussion

We conducted a large-scale genetic analysis to understand the role of cigarette smoking and alcohol consumption with ACE2 expression in multiple tissues/organs, comprehending its role in the prevention of COVID-19. Strong IVs were constructed using hundreds of SNPs associated with smoking and alcohol consumption. We capitalized on the summary statistics of the largest tissue-specific eQTL conducted for ACE2 expression levels and the most up-to-date GWAS data of COVID-19 related adverse outcomes. We found a putative causal relationship between smoking-related phenotypes and an increased ACE2 expression in multiple tissues, as well as an increased susceptibility and severity of COVID-19.

Our findings are supported by previous epidemiological studies which have demonstrated a significant association between smoking and COVID-19 disease progression or death (*Guan et al., 2020*; *Hu et al., 2020*; *Mehra et al., 2020*). For example, a study involving 214 patients with laboratory confirmed COVID-19 from Wenzhou China found that compared to non-severe cases, patients with severe disease were more likely to be smokers (26.3% vs. 6.4%, p = 0.038) (*Zheng et al., 2020*). Another study recruiting 78 patients with COVID-19 in Wuhan also found a higher proportion of ever-smokers in COVID-19 progression group than in the improvement/stabilization group (OR=14.28, 95% CI:1.58–25.00, p=0.018) (*Liu et al., 2020*). Consistent with these findings, a meta-analysis on a total of 11,590 COVID-19 cases demonstrated a higher proportion of smokers among 2133 patients experienced disease progression, suggesting that smoking aggravated COVID-19 progression (OR=1.91, 95% CI: 1.42–2.59) (*Patanavanich and Glantz, 2020*). On the contrary, a few small-scale studies with sample sizes ranging from 44 to 191 conducted in Wuhan China did not report remarkable association of smoking with COVID-19 severity or progression (*Huang et al., 2020*; *Yang et al., 2020*; *Zhang et al., 2020*; *Zhou et al., 2020b*). For instance, a retrospective study including 191 patients of which 137 survived and 54 non-survived found a comparable proportion of smokers in survivors and non-survivors (9% vs. 4%, p=0.21) (*Zhou et al., 2020b*). A meta-analysis including five studies (four in Wuhan and one across 30 provinces in Mainland China) (*Guan et al., 2020*; *Liu et al., 2020*; *Huang et al., 2020*; *Yang et al., 2020*; *Zhang et al., 2020*) involving 1399 individuals with COVID-19 revealed no significant association between smoking and disease severity (OR=1.69, 95% CI: 0.41–6.92) (*Lippi and Henry, 2020*). Opposite to the findings from Asian population, evidence from the Veterans Affairs Birth Cohort in a US population found that current smoking was associated with a lower risk of COVID-19 susceptibility (OR=0.45, 95% CI: 0.35–0.57) (*Rentsch et al., 2020*). The contradictory results in those small-scale studies might be due to insufficient power, low proportion of smokers, and a limited representativeness of the study population. For example, compared with the high proportion of smokers in China (an average 26.6% prevalence in the general population), only 1.4% patients were current smokers in the study conducted by *Zhang et al., 2020* and 12.6% in the study conducted by *Guan et al., 2020*. Given these discrepancies, additional studies are warranted to confirm the role of smoking in both the onset and progression of COVID-19.

In addition to clinical observational studies, laboratory examination has demonstrated the importance of tissues specificity. For example, Cai et al. identified that smoking was associated with an elevated expression of ACE2 in the lung, providing biological evidence of smoking with an increased susceptibility to SARS-CoV-2 infection or severity (*Cai, 2020*). Moreover, Rao et al. conducted a phenome-wide MR study incorporating 3948 traits, diseases, and blood proteins and identified a nominal significant association between tobacco use and ACE2 expression in the lung (IVW: β=0.918, p=0.016) (*Rao et al., 2020*). However, this association did not pass multiple comparisons (p-value

for FDR was 0.51), which was consistent with our MR results using a greatly augmented number of IVs (for smoking status IV=378).

The biological mechanisms underlying smoking and tissue-specific ACE2 expression remain to be disclosed. ACE2 has been considered as the target receptor of SARS-CoV-2 entry into the host cells and an increased expression of ACE2 appears to raise both the susceptibility and severity of COVID-19. As a type I transmembrane metallocarboxypeptidase homologous to ACE, ACE2 is known to be expressed in a variety of tissues, including respiratory tract, cardio-renal tissues, and gastrointestinal tissues (*Harmer et al., 2002*). Our study found that smoking increased ACE2 expression in multiple tissues including the brain and colon tissues. While SARS-CoV-2 mainly spreads via respiratory tract, enrichment of SARS-CoV-2 in gastrointestinal tract has been confirmed by testing viral RNA in stool from 71 patients with COVID-19, suggesting the importance of gastrointestinal involvement in the infection (*Xiao et al., 2020*). Furthermore, nearly one-fifth COVID-19 patients remained SARS-CoV-2 RNA-positive in their stool, despite negative results in their respiratory samples. Of those 71 patients, ACE2 was abundantly expressed in the gastrointestinal epithelia, but rarely expressed in the esophageal epithelia. In addition, findings from public databases (GEO, GTEx, and HPA) have demonstrated a higher expression level of ACE2 in the gastrointestinal tract (colon, rectal, and small intestine) and liver than in the lung (*Burgueño et al., 2020*; *Li et al., 2020*; *Pirola and Sookoian, 2020*). Taking these pieces of evidence together, we could reasonably assume that the biological mechanisms underlying the link between ACE2 expression and COVID-19 susceptibility are complicated, involving multiple organs other than the lung. Consistent with our findings on a link between smoking and increased ACE2 expression in both transverse colon and sigmoid colon, these results collectively suggest that smoking mediates ACE2 expression in gastrointestinal tract, subsequently influence the susceptibility and severity of COVID-19.

We also found that smoking influenced ACE2 expression in the brain tissue, especially in putamen basal ganglia and hypothalamus. Smoking may promote cellular uptake of SARS-CoV-2 through nicotinic acetylcholine receptor (nAChR) signalling (*Russo et al., 2020*). It is worth noting that nAChR and ACE2 are known to be co-expressed on many sites such as cortex, striatum, and hypothalamus within human brain (*Dani and Bertrand, 2007*; *Jones et al., 2009*). Nicotine stimulation of the nAChR increased ACE2 expression within neural cells, indicating a likelihood that smokers are more vulnerable to COVID-19 (*Olds and Kabbani, 2020*).

To the best of our knowledge, we performed a large-scale phenome-wide MR study to understand a causal role of smoking and alcohol consumption in ACE2 expression, as well as in COVID-19 related outcomes, totalling 532 genetic associations from 1.2 million individuals of European ancestry and covering almost all tissues/organs of human body (N=44). In addition, we rigorously selected proxy SNPs and performed a series of sensitivity analyses to satisfy MR model assumptions, for example, we satisfied the 'relevance' assumption by using GWAS-identified significant SNPs as IVs; we ensured the 'independent' assumption and the 'exclusion restriction' assumption by performing several important sensitivity analyses. However, limitations need to be acknowledged. Although hundreds of SNPs were used as proxies for cigarette smoking and alcohol consumption, these GWAS-identified SNPs explained only a small fraction (1–2%) of phenotypic variance. In addition, the minimum sample size of 114 in brain substantia nigra tissue further limited the statistical power of MR analysis, albeit GTEx database is so far the largest available database with both genotype and expression data. Moreover, data on the genotypes, tissue expression, and COVID-19 related outcomes of our analyses were all based on European ancestry populations, restricting its generalizability to other ethnicities. On the other hand, our data, with exposure and outcome GWAS(s) (or eQTLs) conducted using individuals of the same underlying populations (all European ancestry), greatly reduces the population stratification as well as satisfies the MR model assumption – for a two-sample MR to be valid, the two samples have to be preferably from the same ethnicity. Finally, we have to acknowledge that few of our significant associations passed the stringent Bonferroni corrections. Our results provide a comprehensive picture for the causal relationship of smoking with ACE2 expression in various tissues and with COVID-19 susceptibility and severity, yet we stress caution for interpretation and extra analyses are needed to replicate these findings.

In conclusion, genetically instrumented smoking phenotypes reflected by both smoking initiation and smoking intensity are significantly associated with a high expression level of ACE2 in multiple tissues/organs, subsequently increasing the susceptibility and severity of COVID-19. Our results provide important clinical implications on that smokers might be more vulnerable to SARS-CoV-2

infection or severe disease. At the population level, smoking cessation is also an important actionable prevention strategy to COVID-19. Further studies are needed to confirm or refute our findings.

## Acknowledgements

We would like to thank the GSCAN consortium, the GTEx Program, and the COVID-19 Host Genetics Initiative for the release of their data.

## Additional information

### Funding

| Funder | Grant reference number | Author |
|---|---|---|
| Swedish Research Council | VR-2018-02247 | Xia Jiang |
| Swedish Research Council for Health, Working Life and Welfare | FORTE-2020-00884 | Xia Jiang |

The funders had no role in study design, data collection and interpretation, or the decision to submit the work for publication.

### Author contributions

Hui Liu, Data curation, Writing - original draft, Writing - review and editing; Junyi Xin, Sheng Cai, Data curation, Formal analysis, Writing - review and editing; Xia Jiang, Conceptualization, Supervision, Writing - original draft, Writing - review and editing

### Author ORCIDs

Hui Liu  https://orcid.org/0000-0002-5531-3640
Junyi Xin  http://orcid.org/0000-0001-6677-3936
Xia Jiang  https://orcid.org/0000-0001-5878-8986

### Decision letter and Author response

Decision letter https://doi.org/10.7554/eLife.64188.sa1
Author response https://doi.org/10.7554/eLife.64188.sa2

## Additional files

### Supplementary files

• Supplementary file 1. Supplementary results of the causality of smoking and alcohol consumption on angiotensin-converting enzyme 2 (ACE2) expression and coronavirus disease 2019 (COVID-19) related outcomes. (a) Information of genome-wide significant single nucleotide polymorphisms (SNPs) associated with smoking initiation. (b) Information of genome-wide significant SNPs associated with cigarettes per day. (c) Information of genome-wide significant SNPs associated with drinks per week. (d) Sample size of ACE2 expression in 44 tissues/organs from GTEx database. (e) Statistical power in Mendelian randomization study on causal association of smoking and alcohol consumption with ACE2 expression and COVID-19 related adverse outcomes. (f) Causal association of smoking initiation and ACE2 expression. (g) Causal association of cigarettes per day and ACE2 expression. (h) Causal association of drinks per week and ACE2 expression. (i) Causal association of smoking initiation and ACE2 expression excluding palindromic SNPs. (j) Causal association of cigarettes per day and ACE2 expression excluding palindromic SNPs. (k) Causal association of drinks per week and ACE2 expression excluding palindromic SNPs. (l) Causal association of smoking initiation and ACE2 expression excluding pleiotropic SNPs. (m) Causal association of cigarettes per day and ACE2 expression excluding pleiotropic SNPs. (n) Causal association of drinks per week and ACE2 expression excluding pleiotropic SNPs. (o) Causal association of cigarettes per day with the risk of COVID-19 related adverse outcomes. (p) Causal association of drinks per week with the risk of COVID-19 related adverse outcomes. (q) Causal association of smoking and alcohol consumption

with the risk of COVID-19 related adverse outcomes excluding palindromic SNPs. (r) Causal association of smoking and alcohol consumption with the risk of COVID-19 related adverse outcomes excluding pleiotropic SNPs.

• Transparent reporting form

### Data availability

All data generated or analysed during this study are publicly-available. Data of the GSCAN consortium, the GTEx project, and the COVID-19 Host Genetics Initiative can be acessed at https://genome.psych.umn.edu/index.php/GSCAN, https://www.gtexportal.org, and https://www.cov-id19hg.org, respectively. Furthermore, data and main programming codes with annotations have been uploaded to GitHub and made publicly available at https://github.com/hye-hz/MR_Smoke_COVID19.git (copy archived at https://archive.softwareheritage.org/swh:1:rev:1a2038517d8f2c7c772e69c9c5abab7713add9bb).

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

## Appendix 1

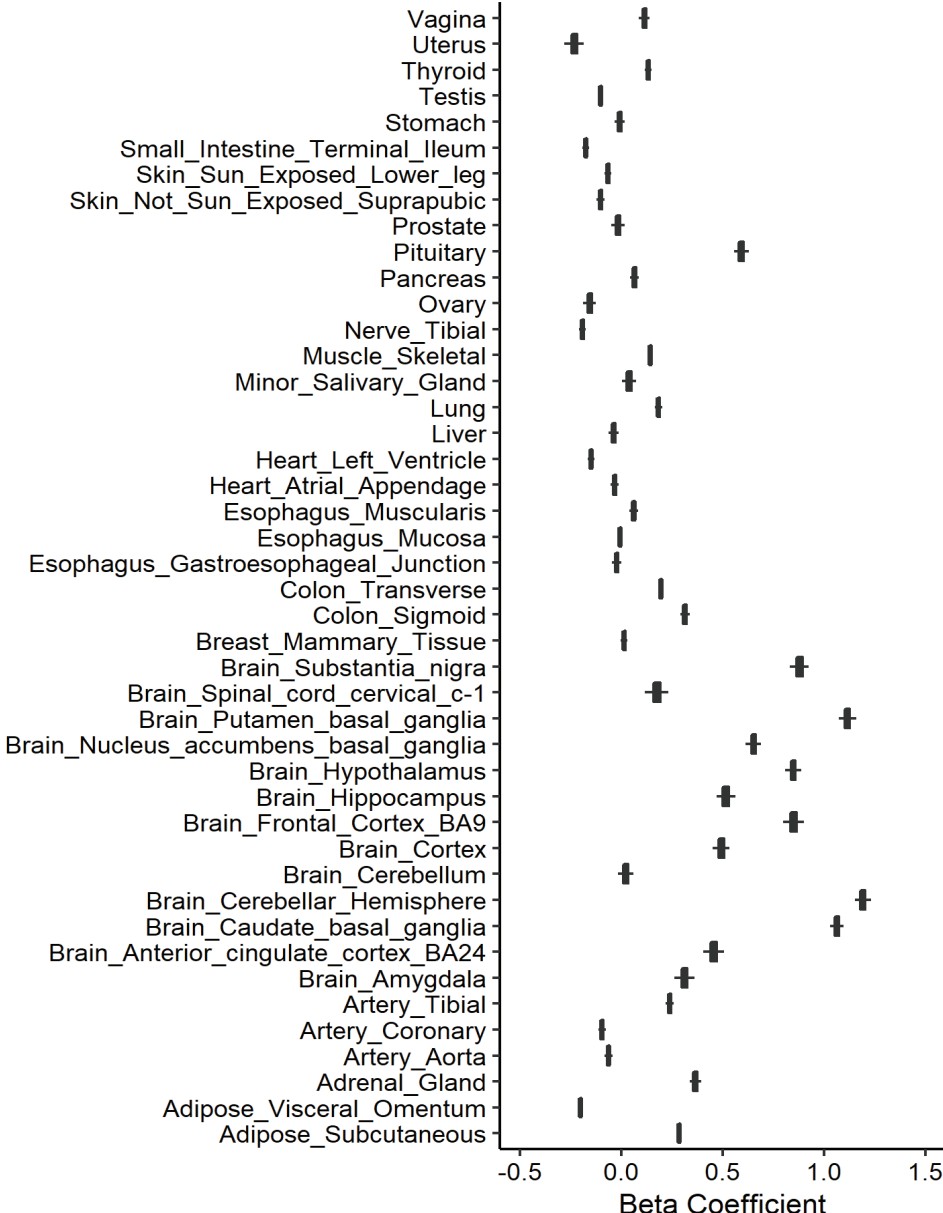

**Appendix 1—figure 1.** Leave-one-out analysis of causal association of smoking initiation with angiotensin-converting enzyme 2 (ACE2) expression. Each boxplot represents the centralized tendency of effect sizes (beta coefficients) of smoking initiation on tissue-specific ACE2 expression based on the results of leave-one-out analysis where we excluded one single nucleotide polymorphism (SNP) at a time and performed inverse-variance weighted (IVW) using the remaining SNPs.

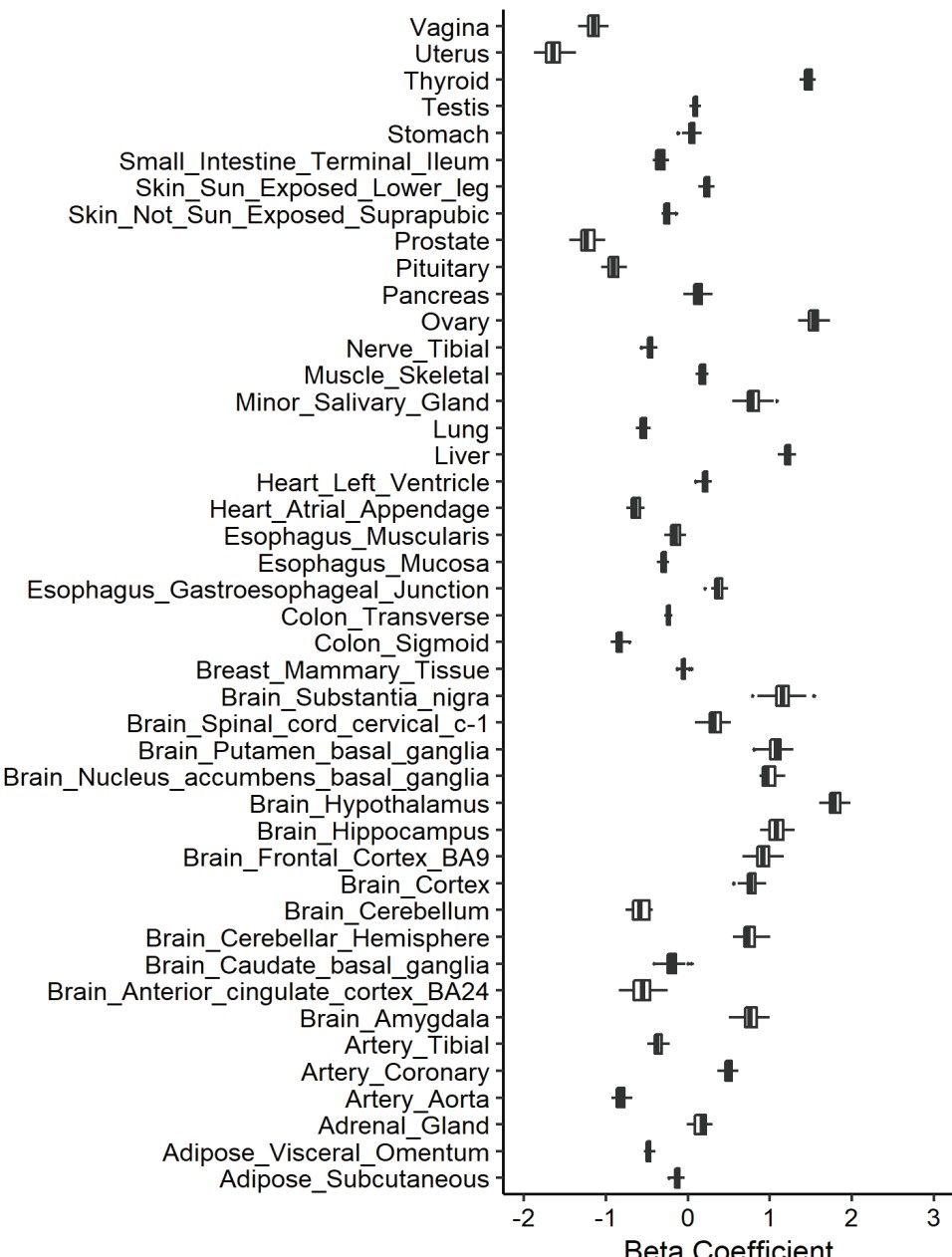

**Appendix 1—figure 2.** Leave-one-out analysis of causal association of cigarettes per day with angiotensin-converting enzyme 2 (ACE2) expression. Each boxplot represents the centralized tendency of effect sizes (beta coefficients) of cigarettes per day on tissue-specific ACE2 expression based on the results of leave-one-out analysis where we excluded one single nucleotide polymorphism (SNP) at a time and performed inverse-variance weighted (IVW) using the remaining SNPs.

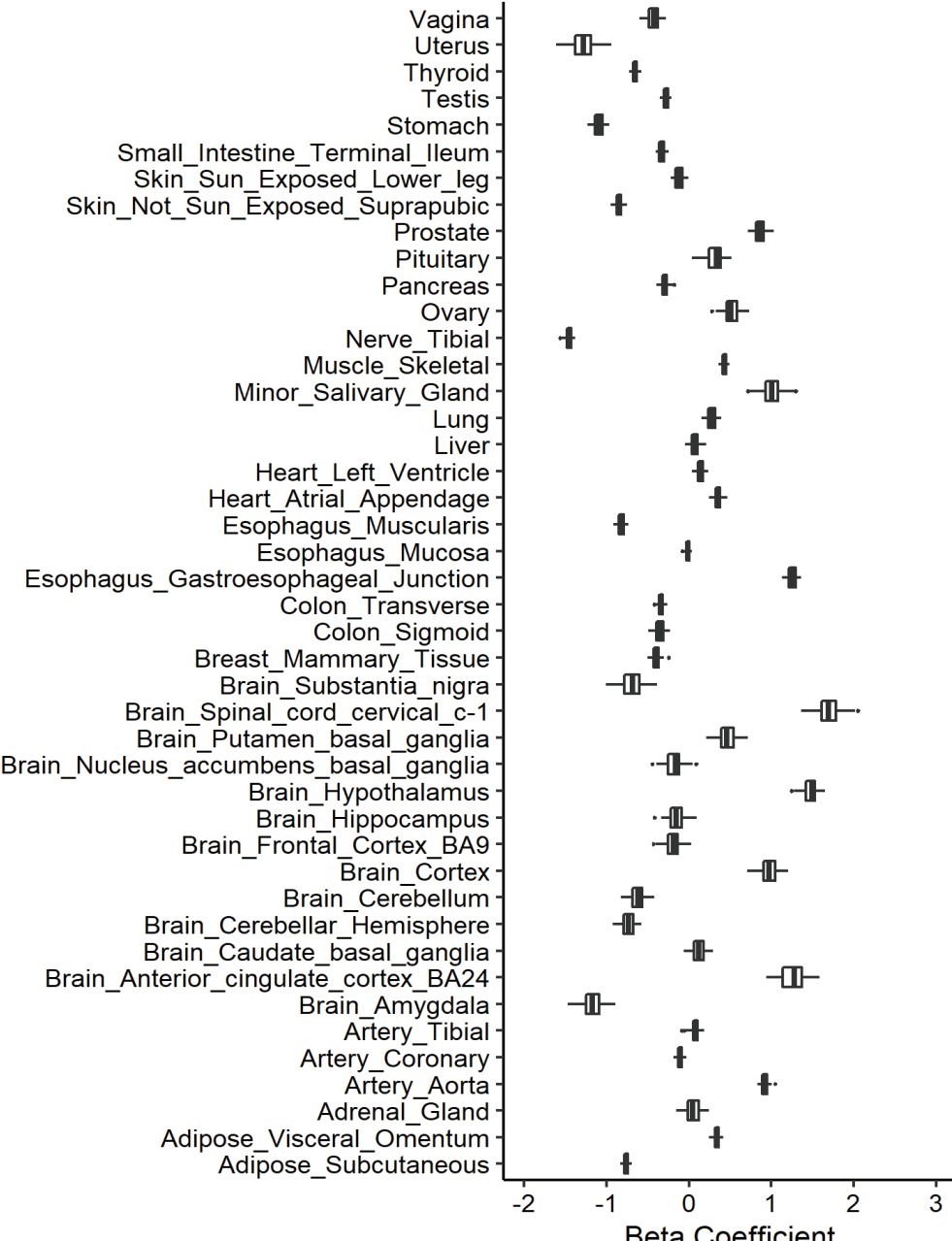

**Appendix 1—figure 3.** Leave-one-out analysis of causal association of drinks per week with angiotensin-converting enzyme 2 (ACE2) expression. Each boxplot represents the centralized tendency of effect sizes (beta coefficients) of drinks per week on tissue-specific ACE2 expression based on the results of leave-one-out analysis where we excluded one single nucleotide polymorphism (SNP) at a time and performed inverse-variance weighted (IVW) using the remaining SNPs.

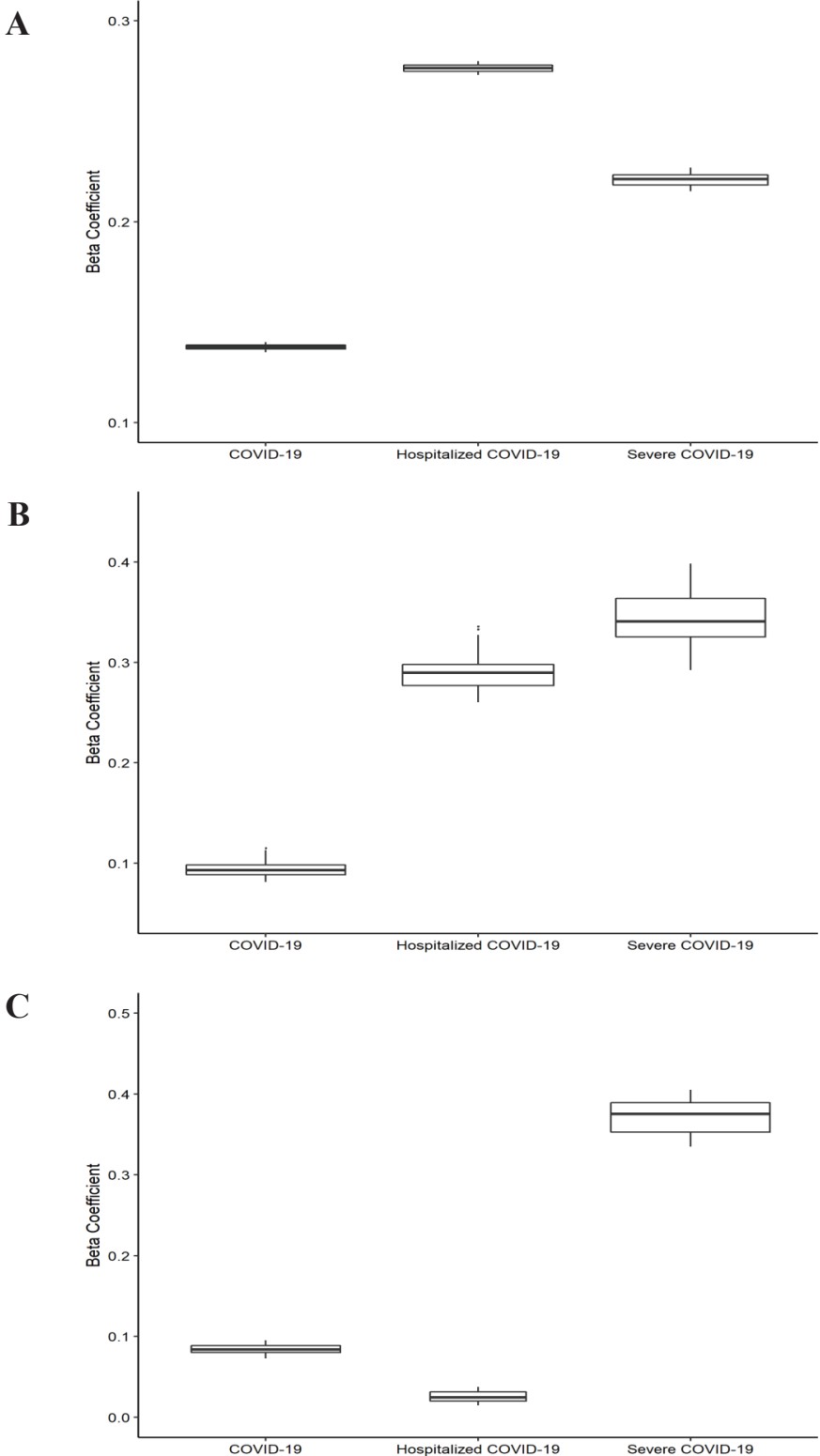

**Appendix 1—figure 4.** Leave-one-out analysis of causal association of smoking and alcohol consumption with coronavirus disease 2019 (COVID-19) related adverse outcomes. (**A**) Leave-one-out analysis of causal association of smoking initiation with COVID-19 related adverse outcomes. (**B**)

*Appendix 1—figure 4 continued on next page*

*Appendix 1—figure 4 continued*

Leave-one-out analysis of causal association of cigarettes per day with COVID-19 related adverse outcomes. (**C**) Leave-one-out analysis of causal association of drinks per week with COVID-19 related adverse outcomes. Each boxplot represents the centralized tendency of effect sizes (beta coefficients) of exposure (including smoking and alcohol consumption) on COVID-19 related adverse outcomes based on the results of leave-one-out analysis where we excluded one single nucleotide polymorphism (SNP) at a time and performed inverse-variance weighted (IVW) using the remaining SNPs. COVID-19 indicates susceptibility to COVID-19. Severe COVID-19 indicates very severe respiratory confirmed COVID-19.

