## [Decision Letter]

**Acceptance summary:**

This work presents a two-sample Mendelian Randomization (MR) analysis of smoking and alcohol consumption with ACE2 expression in multiple organs. The MR approach allows to explore a causal role of these modifiable lifestyle factors in ACE2 expression in 44 tissues/organs using data from the GCSCAN consortium and GTEx. The MR analysis finds interesting associations with smoking status and intensity and increased levels of ACE2 expression in organs that may go on to modify susceptibility to COVID-19. However, no evidence for an effect of alcohol was seen.

**Decision letter after peer review:**

Thank you for submitting your article "Mendelian randomization analysis provides causality of smoking on the expression of ACE2, a putative SARS-CoV-2 receptor" for consideration by *eLife*. Your article has been reviewed by 2 peer reviewers, and the evaluation has been overseen by a Reviewing Editor and a Senior Editor. The following individuals involved in review of your submission have agreed to reveal their identity: Houfeng Zheng (Reviewer #1); Derrick Bennett (Reviewer #2).

The reviewers have discussed the reviews with one another and the Reviewing Editor has drafted this decision to help you prepare a revised submission.

We would like to draw your attention to changes in our policy on revisions we have made in response to COVID-19 (https://elifesciences.org/articles/57162). Specifically, when editors judge that a submitted work as a whole belongs in eLife but that some conclusions require a modest amount of additional new data, as they do with your paper, we are asking that the manuscript be revised to either limit claims to those supported by data in hand, or to explicitly state that the relevant conclusions require additional supporting data. Our expectation is that, where possible, the authors will eventually carry out the additional work and report on how they affect the relevant conclusions either in a preprint on bioRxiv or medRxiv, or if appropriate, as a Research Advance in eLife, either of which would be linked to the original paper.

Summary:

This paper presents a two-sample Mendelian Randomization analysis of smoking and alcohol consumption with ACE2 expression in multiple organs using tissue samples made available through the GTEx dataset. The MR analysis found promising associations with smoking status and intensity and increased levels of ACE2 expression in organs that may go on to modify susceptibility to COVID-19. While the research is novel, the methods have been applied to only one data resource and some of the conclusions are not warranted by the data and analysis. In particular the causal inferences relating to COVID-19 susceptibility require additional data. Many conclusions are too strong based on the analyses performed.

Essential revisions:

1. The authors must include a detailed flowchart of how SNPs were selected/excluded for each IV. The information must be reported in sufficient detail so that the IVs could be recreated and the whole MR analysis could be replicated. If this requires substantial programming the annotated code could be made available through GitHub for example.

2. The authors report "Our results provide important clinical implications on that smokers might be more vulnerable to SARS-CoV-2 infection or severe disease." They have not directly assessed the relationship of their IV for smoking to SARS-CoV-2 infection so their conclusions need to be toned down. Did the authors consider obtaining SARS-COV-2 outcome data from the COVID-19 Host Genetics Initiative (https://doi.org/10.1038/s41431-020-0636-6)? This would greatly strengthen the report.

3. The MR studies have been conducted in a European population so are these results generalizable to other populations? While the resource used is impressive can this analysis be replicated in an independent data set? Even if some of the signals could be replicated this would add enormous value to the results.

4. Can you offer an explanation why ACE2 was highly expressed on brain, colon, liver et al., but not on respiratory tract and lung tissue? Is higher expression of ACE2 really susceptible factor for Covid19? What is the evidence?

5. The Discussion section seems to focus on evidence from China but the results from China may be affected by the sex-differences in smoking and alcohol prevalence. The patterns of smoking and alcohol in East Asian populations is very different from Western populations. Typically very few women smoke or drink in East Asia. The authors should comment on this.

6. Did the authors consider performing analyses separately for men and women in this study?

7. Table 2 shows the detectable difference with a fixed power of 80% and a significance level of 0.05. Should the significance level be modified to deal with multiple testing with sample from different organs? If not why not? The sample size calculation requires further clarification. It is preferable to mention the a priori power calculations in the methods section of the report not the results. Are the effect sizes detectable clinically relevant? How was this ascertained?

8. The authors need to report the associated F-statistics for their instrumental variables.

9. The authors mention that "Expression values for each gene were inverse quantile normalized to a standard normal distribution across samples". So this suggests that the results are based on a per standard deviation change but this is not clear from the results.

10. Only MR-Egger was used to assess horizontal pleiotropy. There are several other approaches that make different assumptions to MR-Egger that should be considered in order to triangulate the findings.

11. In the MR results, the significance from IVW approach were not replicated in MR-Egger regression, and vice versa. Can we believe these are real casual associations? Could you explain?

12. The abstract should communicate the size of the dataset (ie the number of samples) and report an effect size, 95% confidence interval and p-value for each signal reported in the abstract. Stating 'significant' or 'non-significant' is not appropriate for an abstract.

13. In Figures 1, 2 and 3 the x-axis needs clearer labelling. Isn't this a plot of β values per 1 SD change in ACE2 expression?

---

## [Author Response]

Summary:This paper presents a two-sample Mendelian Randomization analysis of smoking and alcohol consumption with ACE2 expression in multiple organs using tissue samples made available through the GTEx dataset. The MR analysis found promising associations with smoking status and intensity and increased levels of ACE2 expression in organs that may go on to modify susceptibility to COVID-19. While the research is novel, the methods have been applied to only one data resource and some of the conclusions are not warranted by the data and analysis. In particular the causal inferences relating to COVID-19 susceptibility require additional data. Many conclusions are too strong based on the analyses performed.

We thank the reviewer for a nice summary of our paper as well as for pointing out the potential limitations, all of which are very solid, and we totally agree with. In our revised manuscript, we have added extra analyses leveraging GWAS summary statistics of COVID-19 related outcomes to bring in an additional source of data. We have also revised the whole manuscript to avoid a possible over-selling of results. We feel that our manuscript has been greatly improved and hope it now meets the criteria for publication in *eLife*.

Essential revisions:1. The authors must include a detailed flowchart of how SNPs were selected/excluded for each IV. The information must be reported in sufficient detail so that the IVs could be recreated and the whole MR analysis could be replicated. If this requires substantial programming the annotated code could be made available through GitHub for example.

We thank the reviewer for pointing this issue out. Indeed, selection of IVs serves as an important basis for MR analysis.

Following with the reviewer’s suggestion, we have added a flowchart of IV-selection as Figure 1. Specific information regarding each IV including its rsID, genomic coordinates, effect size, allele frequency etc. can be found in our Supplementary File 1a-1c. Data and main programming codes with annotations have been uploaded to GitHub and made publicly available at https://github.com/hye-hz/MR_Smoke_COVID19.git.

2. The authors report "Our results provide important clinical implications on that smokers might be more vulnerable to SARS-CoV-2 infection or severe disease." They have not directly assessed the relationship of their IV for smoking to SARS-CoV-2 infection so their conclusions need to be toned down. Did the authors consider obtaining SARS-COV-2 outcome data from the COVID-19 Host Genetics Initiative (https://doi.org/10.1038/s41431-020-0636-6)? This would greatly strengthen the report.

We thank the reviewer for this extremely constructive comment as well as for pointing us to the COVID-19 outcome-related GWAS data. Incorporating those data will greatly enhance the validity of our current results. Following with the reviewer’s suggestions, we have estimated the putative causal effect of smoking and alcohol consumption on COVID-19 susceptibility and severity. Consistent with our ACE2 expression results, both smoking initiation and intensity were associated with an increased risk of COVID-19 related outcomes while alcohol consumption did not seem to influence the risk.

We have formulated those results into extra tables (Table 3 and Supplementary File 1o-1r) and added to our manuscript, the relevant texts read:

“Finally, complementing to findings of ACE2 expression, we tested a putative causal link between smoking status, smoking intensity, alcohol consumption and the risk of COVID-19 related adverse outcomes. We found that smoking initiation significantly increased the risk of COVID-19 onset (IVW: OR=1.15, 95%CI: 1.07-1.23, *p*=8.7×10^-5^) even after taking into account multiple comparisons (Table 3). Results remained significant in the weighted median and MR-PRESSO methods, however, showed larger statistical uncertainties in the MR-Egger regression. In addition, smoking initiation increased the risk of very severe respiratory confirmed COVID-19 and hospitalized COVID-19 using the IVW approach, but the results were not supported by the MR-Egger regression. Smoking intensity (cigarettes per day) only increased the risk of very severe respiratory confirmed COVID-19 as shown in the MR-Egger regression (OR=5.99, 95%CI: 1.57-22.84, *p*=0.012) (Supplementary File 1o). On the contrary, we did not find a causal link between alcohol consumption (drinks per week) and the risk of COVID-19 adverse outcomes (Supplementary File 1p). Findings did not alter in the sensitivity analyses (Supplementary File 1q-1r and Appendix 1-Figure 4)”

3. The MR studies have been conducted in a European population so are these results generalizable to other populations? While the resource used is impressive can this analysis be replicated in an independent data set? Even if some of the signals could be replicated this would add enormous value to the results.

We thank the reviewer for raising such an important point regarding the generalizability of results. Indeed, only data of European ancestry populations were used in our study and therefore our results had limited generalizability to other ethnicities. We have acknowledged this limitation in our Discussion, it reads:

“Moreover, data on the genotypes, tissue expression and COVID-19 related outcomes of our analyses were all based on European ancestry populations, restricting its generalizability to other ethnicities. On the other hand, our data, with exposure and outcome GWAS(s) (or eQTLs) conducted using individuals of the same underlying populations (all European ancestry) greatly reduces the population stratification as well as satisfies the MR model assumption – for a two-sample MR to be valid, the two samples have to be preferably from the same ethnicity.”

Regarding the replication of results, we unfortunately couldn’t find proper GWAS conducted for our exposures of interest (smoking, alcohol consumption) or outcomes (ACE2 expression, COVID-19 adverse outcomes) in populations other than in the European ancestry populations. Following with the reviewer’s suggestion, we have incorporated COVID-19 outcome-related GWAS which serve as an additional source of data and corroborate with our ACE2 expression findings to add values to our original results. Please read our response to question #2.

4. Can you offer an explanation why ACE2 was highly expressed on brain, colon, liver et al., but not on respiratory tract and lung tissue? Is higher expression of ACE2 really susceptible factor for Covid19? What is the evidence?

We shared the same concerns as the reviewer regarding the interpretation of our results. We tried to offer explanation from three aspects as shown below. The reviewer is very welcome to provide further comments and/or suggestions based on our explanations.

First of all, our results were consistent with the study conducted by Rao et al. which examined the effect of an extensive amount of diseases, traits and blood proteins (N=3948) on ACE2 expression in the lung^1^. For smoking, 3 SNPs associated with tobacco use were extracted from the UK Biobank (ukb-b-5115) and used as IVs; Data on ACE2 expression of the lung tissue were extracted from the GTEx database and used as the outcome. Despite a nominal significant association of tobacco use and ACE2 expression in the lung (IVW: β=0.92, SE=0.38, *p*=0.016), this result did not pass multiple comparisons (*p* value for FDR was 0.51). Our results, with 100 times augmented numbers of IVs and largely increased statistical power, did not find a causal relationship between genetically predicted smoking and ACE2 expression in the lung.

Secondly, as we mentioned on Page 9-10 in our Discussion section:

“While SARS-CoV-2 mainly spreads via respiratory tract, enrichment of SARS-CoV-2 in gastrointestinal tract has been confirmed by testing viral RNA in stool from 71 patients with COVID-19, suggesting the importance of gastrointestinal involvement in the infection^2^. Furthermore, nearly one fifth of COVID-19 patients remained SARS-CoV-2 RNA positive in their stool, despite their negative results in their respiratory samples. Of those 71 patients, ACE2 was abundantly expressed in the gastrointestinal epithelia, but rarely expressed in the esophageal epithelium. In addition, findings from public databases (GEO, GTEx and HPA) have demonstrated a higher expression level of ACE2 in the gastrointestinal tract (colon, rectal, and small intestine) and liver than in the lung^3-5^. Taking these pieces of evidence together, we could reasonably assume that the biological mechanisms underlying the link between ACE2 expression and COVID-19 susceptibility is complicated, involving multiple organs other than the lung.”

Last but not the least, as the reviewer mentioned in his original question – “Is higher expression of ACE2 really a susceptible factor for COVID-19? what is the evidence?” – We have to admit that there is a lack of strong evidence supporting an absolute causal role of ACE2 in the susceptibility of COVID-19. This infection, as all other complex traits or diseases, occurs under a multifactorial etiology with the involvement of multiple factors, such as a non-trivial role of environmental exposures (here, virus load for example) and various immunological / physiological molecular events other than the expression of ACE2 receptor. Our study, which focused on ACE2 expression with reasonable hypothesis (a most relevant receptor for the virus) and availability of data, contributes as a preliminary first step to understand the etiology of COVID-19. Downstream experimental data and future large-scale analysis are needed to dispute or support our findings, as well as to explore molecules other than ACE2 receptor.

5. The Discussion section seems to focus on evidence from China but the results from China may be affected by the sex-differences in smoking and alcohol prevalence. The patterns of smoking and alcohol in East Asian populations is very different from Western populations. Typically very few women smoke or drink in East Asia. The authors should comment on this.

We agree with the reviewer, indeed, the patterns of smoking and alcohol consumption in East Asian populations differ from that of Western populations, particularly among women. Since the prevalence of smoking among COVID-19 patients are in general low (range from 1.4% to 21.9%), and the patient population consists mainly males (range from 50% to 73%), the sex difference may impose a minimal effect on the outcome. Nevertheless, following with the reviewer’s suggestion, we have cited results from Western populations in our Discussion section, it reads:

“Opposite to the findings from Asian population, evidence from the Veterans Affairs Birth Cohort in a United States population found that current smoking was associated with a lower risk of COVID-19 susceptibility (OR=0.45, 95%CI: 0.35-0.57)^6^.”

6. Did the authors consider performing analyses separately for men and women in this study?

Unfortunately, we were unable to perform such an analysis. Although most current GWAS(s) were carried out including both men and women, stratified analysis was still performed only to a small extent. None of our exposure or outcome GWAS data (both ACE2 expression and COVID-19 adverse outcomes) were stratified by sex. This analysis could be an important future direction to be focused on when data becomes available.

7. Table 2 shows the detectable difference with a fixed power of 80% and a significance level of 0.05. Should the significance level be modified to deal with multiple testing with sample from different organs? If not why not? The sample size calculation requires further clarification. It is preferable to mention the a priori power calculations in the methods section of the report not the results. Are the effect sizes detectable clinically relevant? How was this ascertained?

We thank the reviewer for covering this aspect. Indeed, Bonferroni correction should be used to take into consideration multiple comparisons. We set our corrected *p* threshold as dividing 0.05 by the number of outcomes 0.05(44+3)=1.0×10^-3^ (including 44 tissues / organs and three COVID-19 related adverse outcomes).

However, as 1) the main aim of this work was to evaluate the evidence of a putative causal role of smoking and alcohol consumption on ACE2 expression levels across a range of human tissues / organs (N=44) rather than to identify a specific tissue; and 2) tissues and organs are not totally independent of each other; we therefore applied a marginal significant level of *p* < 0.05 in our study. For all our results, we reported clearly the *p* to ensure an accurate interpretation. We added explanatory sentences on the reasons of using a marginal significant p value as well as highlighted all the results that passed Bonferroni correction. For example, we found that smoking initiation significantly augmented the risk of COVID-19 susceptibility (IVW: OR=1.15, 95%CI: 1.07-1.23, p=8.7×10^-5^) even after taking into account multiple comparisons, indicating that smokers had a higher risk of developing COVID-19 compared with non-smokers.

Following with the reviewer’s suggestion, we have moved the sample size calculation to the Method section and added additional explanatory sentences, it reads:

“To guarantee statistical power, we only included tissues/organs with at least 100 samples in the GTEx database. Under the current sample size, given 1– 2% of the phenotypic variance of smoking and alcohol consumption explained by IVs, our study had sufficient power (>80%) to detect a causal effect of 0.74 to 2.66 in ACE2 expression, and to detect an OR ranging from 1.11 to 1.39 for COVID-19 related outcomes (Supplementary File 1e).”

Lastly, as the reviewer mentioned in his original question – “Are the effect sizes detectable clinically relevant? How was this ascertained?” – We need to admit that effect size in itself cannot give a clear interpretation of the clinical relevance. Specifically, Dr. Burgess underscores that for a binary exposure (e.g. smoking initiation), MR is most suitable for testing a causal link (if there is a causal relationship) rather than for calculating a causal estimation (how strong is the magnitude of causal relationship and how much risk of outcome can be reduced if this exposure is “blocked”)^7^. Because the causal estimation of a binary exposure assumes that the casual effect is a stepwise function at the point of dichotomization, however, MR estimations perform parametric assumptions. Caution is needed when inferring the causal estimation to clinical relevance using a binary exposure.

8. The authors need to report the associated F-statistics for their instrumental variables.

We thank the reviewer for covering this aspect. Strong instrumental variable is the basic requirement to ensure a valid MR result. Following with the reviewer’s suggestion, the strength of instrumental variable was verified by calculating F-statistics using the formula F=R2(n−1−k)(1−R2)k, where R^2^ is the proportion of variance explained by the instrumental variable, k refers to the number of IVs, and n indicates the sample size^8^. The F-statistics for smoking initiation, smoking intensity (cigarettes per day) and alcohol consumption (drinks per week) were 77.2, 67.4, and 17.8, respectively, indicating strong IVs (F-statistics > 10) for each of our exposure of interest.

9. The authors mention that "Expression values for each gene were inverse quantile normalized to a standard normal distribution across samples". So this suggests that the results are based on a per standard deviation change but this is not clear from the results.

We thank the opportunity for being able to make further clarifications. When saying “Expression values for each gene were inverse quantile normalized to a standard normal distribution across samples”, it referred to the ACE2 expression level (here our outcome). For our exposure, three phenotypes were included in the analyses, that is, smoking initiation (ever vs. never), smoking intensity (cigarettes per day) and alcohol consumption (drinks per week). For the latter two exposures, we indeed calculated results based on a per-SD change of exposure, while for smoking initiation which was a binary exposure, we calculated results based on per-unit change in the exposure on the log odds scale.

10. Only MR-Egger was used to assess horizontal pleiotropy. There are several other approaches that make different assumptions to MR-Egger that should be considered in order to triangulate the findings.

Following with the reviewer’s suggestion, we have tested horizontal pleiotropy using the MR-PRESSO approach, results of which have been added to Tables 1-3, to Supplementary File 1f-1h, and to Supplementary File 1o-1p. There was no evidence for the existence of horizontal pleiotropy according to the global test (all p>1.0×10^-6^).

11. In the MR results, the significance from IVW approach were not replicated in MR-Egger regression, and vice versa. Can we believe these are real casual associations? Could you explain?

We appreciated this opportunity for making further clarifications.

First of all, it is not surprising that the significance from IVW were not replicated in the MR-Egger regression as this method provides twice as large standard errors as IVW and therefore a wider 95% confidence intervals.

Secondly, as we mentioned in our Methods section, “Results were considered significant only if they passed statistical significance (p<0.05) in the IVW approach or the MR-Egger regression and remained directional consistent in the weighted median and MR-PRESSO methods across both primary and sensitivity analyses.”. The IVW approach is the most conventional method which was applied as the primary method to estimate a causal link between exposures (smoking and alcohol consumption) and outcomes (ACE2 expression and COVID-19 related outcome), while MR-Egger regression and several other approaches were used mainly to identify potential horizontal pleiotropy.

12. The abstract should communicate the size of the dataset (ie the number of samples) and report an effect size, 95% confidence interval and p-value for each signal reported in the abstract. Stating 'significant' or 'non-significant' is not appropriate for an abstract.

We thank this constructive suggestion from reviewers and have comprehensively revised our abstract, it reads:

“Background: To understand a causal role of modifiable lifestyle factors in ACE2 expression (a putative SARS-CoV-2 receptor) across 44 human tissues/organs, and in COVID-19 susceptibility and severity, we conducted a phenome-wide two-sample Mendelian randomization (MR) study.

Methods: More than 500 genetic variants were used as instrumental variables to predict smoking and alcohol consumption. Inverse-variance weighted approach was adopted as the primary method to estimate a causal association, while MR-Egger regression, weighted median and MR-PRESSO were performed to identify potential horizontal pleiotropy.

Results: We found that genetically predicted smoking intensity significantly increased ACE2 expression in thyroid (β=1.468, p=1.8×10^-8^); and increased ACE2 expression in adipose, brain, colon and liver with nominal significance. Additionally, genetically predicted smoking initiation significantly increased the risk of COVID-19 onset (odds ratio=1.14, p=8.7×10^-5^). No statistically significant result was observed for alcohol consumption.

Conclusions: Our work demonstrates an important role of smoking, measured by both status and intensity, in the susceptibility to COVID-19.”

13. In Figures 1, 2 and 3 the x-axis needs clearer labelling. Isn't this a plot of β values per 1 SD change in ACE2 expression?

Following with the reviewer’s suggestion, we have revised the footnotes of each figure accordingly.

Taking Appendix 1-Figure 1 as an example, the footnote was stated as “Each boxplot represents the centralized tendency of effect sizes (β coefficients) of smoking initiation on tissue-specific ACE2 expression based on the results of leave-one-out analysis where we excluded one SNP at a time and performed IVW using the remaining SNPs”.

These boxplots (or the leave-one-out analysis) aim to identify outlying SNPs which could potentially bias our causal estimates. For example, if the result was driven by one or a few SNPs with larger effects, then we would expect a drastic change in the estimate when that particular outlying SNP was removed. In our case, all estimates centred around the expected value (from the main analysis using all IVs) indicating an absence of outlying SNPs.

References:

1. Rao S, Lau A, So HC. Exploring Diseases/Traits and Blood Proteins Causally Related to Expression of ACE2, the Putative Receptor of SARS-CoV-2: A Mendelian Randomization Analysis Highlights Tentative Relevance of Diabetes-Related Traits. *Diabetes Care* 2020;**43**(7):1416-1426.

2. Xiao F, Tang M, Zheng X, Liu Y, Li X, Shan H. Evidence for Gastrointestinal Infection of SARS-CoV-2. *Gastroenterology* 2020;**158**(6):1831-1833 e3.

3. Pirola CJ, Sookoian S. COVID-19 and ACE2 in the Liver and Gastrointestinal Tract: Putative Biological Explanations of Sexual Dimorphism. *Gastroenterology* 2020;**159**(4):1620-1621.

4. Li MY, Li L, Zhang Y, Wang XS. Expression of the SARS-CoV-2 cell receptor gene ACE2 in a wide variety of human tissues. *Infect Dis Poverty* 2020;**9**(1):45.

5. Burgueno JF, Reich A, Hazime H, Quintero MA, Fernandez I, Fritsch J, Santander AM, Brito N, Damas OM, Deshpande A, Kerman DH, Zhang L, Gao Z, Ban Y, Wang L, Pignac-Kobinger J, Abreu MT. Expression of SARS-CoV-2 Entry Molecules ACE2 and TMPRSS2 in the Gut of Patients With IBD. *Inflamm Bowel Dis* 2020;**26**(6):797-808.

6. Rentsch CT, Kidwai-Khan F, Tate JP, Park LS, King JT, Skanderson M, Hauser RG, Schultze A, Jarvis CI, Holodniy M, Re VL, Akgün KM, Crothers K, Taddei TH, Freiberg MS, Justice AC. Covid-19 Testing, Hospital Admission, and Intensive Care Among 2,026,227 United States Veterans Aged 54-75 Years. *medRxiv* 2020:2020.04.09.20059964.

7. Burgess S, Labrecque JA. Mendelian randomization with a binary exposure variable: interpretation and presentation of causal estimates. *Eur J Epidemiol* 2018;**33**(10):947-952.

8. Pierce BL, Ahsan H, Vanderweele TJ. Power and instrument strength requirements for Mendelian randomization studies using multiple genetic variants. *Int J Epidemiol* 2011;**40**(3):740-52.

9. Bowden J, Davey Smith G, Haycock PC, Burgess S. Consistent Estimation in Mendelian Randomization with Some Invalid Instruments Using a Weighted Median Estimator. *Genet Epidemiol* 2016;**40**(4):304-14.

10. Verbanck M, Chen CY, Neale B, Do R. Detection of widespread horizontal pleiotropy in causal relationships inferred from Mendelian randomization between complex traits and diseases. *Nat Genet* 2018;**50**(5):693-698.